# Trimetazidine Improves Mitochondrial Dysfunction in SOD1^G93A^ Cellular Models of Amyotrophic Lateral Sclerosis through Autophagy Activation

**DOI:** 10.3390/ijms25063251

**Published:** 2024-03-13

**Authors:** Illari Salvatori, Valentina Nesci, Alida Spalloni, Veronica Marabitti, Maurizio Muzzi, Henri Zenuni, Silvia Scaricamazza, Marco Rosina, Gianmarco Fenili, Mariangela Goglia, Laura Boffa, Roberto Massa, Sandra Moreno, Nicola Biagio Mercuri, Francesca Nazio, Patrizia Longone, Alberto Ferri, Cristiana Valle

**Affiliations:** 1Department of Experimental Medicine, University of Roma “La Sapienza”, 00185 Rome, Italy; illarisalvatori@libero.it; 2IRCCS, Fondazione Santa Lucia, 00179 Rome, Italy; valenesci@gmail.com (V.N.); a.spalloni@hsantalucia.it (A.S.); silviascaricamazza@gmail.com (S.S.); marco.rosina90@gmail.com (M.R.); mercurin@med.uniroma2.it (N.B.M.); p.longone@hsantalucia.it (P.L.); 3Department of Systems Medicine, Tor Vergata University of Rome, 00133 Rome, Italy; 4Department of Biology, Tor Vergata University of Rome, 00133 Rome, Italy; veronicamarabitti1@gmail.com (V.M.); francesca.nazio@uniroma2.it (F.N.); 5Department of Science, LIME, University of Roma Tre, 00165 Rome, Italy; maurizio.muzzi@uniroma3.it (M.M.); sandra.moreno@uniroma3.it (S.M.); 6Unit of Neurology, Fondazione PTV Policlinico Tor Vergata, 00133 Rome, Italy; henri.zenuni@students.uniroma2.eu (H.Z.); mariangelagoglia@gmail.com (M.G.); dott.boffalaura@gmail.com (L.B.); massa@uniroma2.it (R.M.); 7Department of Movement, Human and Health Sciences, University of Rome “Foro Italico”, 00135 Rome, Italy; gianmarcofenili@gmail.com; 8Institute of Translational Pharmacology (IFT), National Research Council (CNR), 00133 Rome, Italy

**Keywords:** Amyotrophic Lateral Sclerosis, Trimetazidine, mitochondria, autophagy, spinal and cortical primary cultures, SOD1^G93A^

## Abstract

Amyotrophic Lateral Sclerosis (ALS) is considered the prototype of motor neuron disease, characterized by motor neuron loss and muscle waste. A well-established pathogenic hallmark of ALS is mitochondrial failure, leading to bioenergetic deficits. So far, pharmacological interventions for the disease have proven ineffective. Trimetazidine (TMZ) is described as a metabolic modulator acting on different cellular pathways. Its efficacy in enhancing muscular and cardiovascular performance has been widely described, although its molecular target remains elusive. We addressed the molecular mechanisms underlying TMZ action on neuronal experimental paradigms. To this aim, we treated murine SOD1^G93A^-model-derived primary cultures of cortical and spinal enriched motor neurons, as well as a murine motor-neuron-like cell line overexpressing SOD1^G93A^, with TMZ. We first characterized the bioenergetic profile of the cell cultures, demonstrating significant mitochondrial dysfunction that is reversed by acute TMZ treatments. We then investigated the effect of TMZ in promoting autophagy processes and its impact on mitochondrial morphology. Finally, we demonstrated the effectiveness of TMZ in terms of the mitochondrial functionality of ALS-rpatient-derived peripheral blood mononuclear cells (PBMCs). In summary, our results emphasize the concept that targeting mitochondrial dysfunction may represent an effective therapeutic strategy for ALS. The findings demonstrate that TMZ enhances mitochondrial performance in motor neuron cells by activating autophagy processes, particularly mitophagy. Although further investigations are needed to elucidate the precise molecular pathways involved, these results hold critical implications for the development of more effective and specific derivatives of TMZ for ALS treatment.

## 1. Introduction 

Amyotrophic Lateral Sclerosis (ALS) is a human neurodegenerative disease characterized by the loss of cortical and spinal motor neurons resulting in muscle wasting. Despite a continuous research effort and numerous clinical trials, currently, no effective therapies exist for this pathology [1,2,3,4,5]. Even though ALS is considered the prototype of motor neuron diseases, systemic metabolic dysregulation has also been described in ALS patients, as well as in animal models of the disease [6,7]. Hypermetabolism, leading to a rapid depletion of energy stores, is observed in about one third of patients, who show a decrease in body mass index, often during a premorbid state [8,9,10,11]. Moreover, metabolic dysfunction in ALS patients correlates with a poor prognosis and a faster decline in locomotor performances [12]. The increased energy expenditure generally comes with hyperlipidemia and a decreased capacity for glucose handling. Furthermore, a large body of evidence collected from patients and animal models describes a drop in energy production possibly ascribable to early mitochondrial failures [6,13,14]. In fact, the mitochondrial alterations linked to a failure in glucose metabolism could hinder the motor unit from meeting the actual energy needs, triggering a vicious cycle where the increased utilization of fatty acids as fuel attempts to compensate for the energy deficits [15,16,17]. In this context, boosting mitochondrial metabolism with targeted pharmacological approaches could improve energy failures that appear early in ALS [18,19,20,21]. Therefore, in this clinical framework, drugs that act as metabolic modulators capable of targeting mitochondria and improving energy metabolism could be beneficial for ALS therapy.

Trimetazidine (TMZ) is a clinically effective drug used for the management of coronary diseases by virtue of its boosting action on energy metabolism [22,23].

Nowadays, the mechanism of action of TMZ remains elusive, although it has been reported to be an inhibitor of the long-chain mitochondrial 3-ketoacyl coenzyme A (ACAA2) thiolase, an enzyme that catalyzes the oxidation of the long-chain fatty acids [24,25], and a compound capable of inhibiting the opening of the mitochondrial permeability transition pore [26,27]. Moreover, TMZ is a drug that promotes glucose utilization in ischemic cardiac muscle by improving glucose handling [28,29]. The preferential use of glucose, compared to other sources of energy, is an advantageous cellular strategy that results in the optimization of ATP production since the terminal mitochondrial oxidation of glucose is the reaction with the lowest oxygen–carbon atom ratio [30]. Moreover, the efficacy of TMZ has been tested in murine models of ageing and cancer-related cachexia, where it increased mitochondrial protein levels, mitochondrial membrane potential and, more generally, muscle performances and neuromuscular communications [31,32].

In line with this evidence, we previously reported that the chronic administration of TMZ significantly prolongs life span alongside improving locomotor functions in SOD1^G93A^ mice [33]. In this animal model, the drug leads to the maintenance of the whole motor unit’s integrity, hindering muscle metabolic shift, neuromuscular junction (NMJ) dismantlement, peripheral nerve demyelination, loss of motor neurons and neuroinflammation. Moreover, the drug improves energy metabolism, ameliorating glucose handling and decreasing energy expenditure. At the base of this metabolic rewiring, there is an enhancement of the mitochondrial functions observed both in spinal cord and in skeletal muscle, which is reflected by a net increase in ATP production. 

Overall, this body of evidence points out a critical role of TMZ in boosting mitochondrial metabolism, although, currently, the mechanism of action by which the drug directly or indirectly targets mitochondrial metabolism is still debated. Furthermore, TMZ activity has been studied mainly in cardiac and skeletal muscle, while its mechanism of action on the nervous tissue is still neglected and, indeed, remains quite obscure.

To elucidate the mechanism of action of TMZ against an ALS neuronal background we utilized primary spinal cord and cortical cell (PSC and PCC, respectively) cultures from SOD1^G93A^ mice. These cellular models have been previously employed and have enabled the highlighting of various pathological processes associated with ALS, such as heightened sensitivity to H2S [34] and the toxicity induced by kainate [35], along with aberrant activity in voltage-gated calcium and AMPA receptors [36].

In these neuronal cultures, we reported that mitochondrial functionality is strongly compromised against the ALS background and that the acute administration of TMZ reverts this phenotype, improving respiration parameters, ATP production, electron transport chain (ETC) complex activity and mitochondrial morphology. The improvement in mitochondrial functions and morphology stems from the ability of TMZ treatment to enhance autophagy/autophagy flux. 

## 2. Results 

### 2.1. Mitochondrial Functionality Is Impaired in Primary Spinal Cord and Cortical Cell Cultures

To investigate the molecular pathways underlying the mechanism of action of TMZ in a neuronal context, we utilized PSC and PCC cultures obtained from WT and SOD1^G93A^ mice. 

We performed an in-depth analysis of mitochondrial bioenergetic parameters by employing real-time measurement of oxygen consumption rate (OCR) with Seahorse^®^ XFe Technologies to characterize the bioenergetic profile of these cell cultures. In detail, we analyzed the basal respiration, which represents the oxygen consumption used to meet cellular ATP demand, ATP-linked respiration, the maximal respiration, which represents the maximal oxygen consumption rate attained by adding the uncoupler carbonyl cyanide-p-trifluoromethoxyphenylhydrazone (FCCP), and the spare respiratory capacity, a parameter representing the amount of additional ATP that can be generated by oxidative phosphorylation in the event of a rapid request or energy demand.

Our findings strongly indicate significant mitochondrial impairment in PSCs and PCCs from SOD1^G93A^ mice compared to from control mice (WT) (Figure 1a,b).

Specifically, in both PSCs and PCCs obtained from SOD1^G93A^ mice, we observed significant reductions in all parameters related to mitochondrial respiration (Figure 1c,d). These impairments were more pronounced in PSCs compared to PCCs. Finally, to exclude the possibility that the lower respiratory rates could reflect a lower mitochondrial biomass in the SOD1^G93A^ cells, we evaluated the expression of the mitochondria-specific markers TOM20 and VDAC. The obtained results did not show substantial quantitative differences (Appendix A).

These results highlight that mitochondrial impairment is an intrinsic characteristic of ALS, manifesting early in spinal cord and cortical neurons. Although mitochondrial impairments in the spinal cord of SOD1^G93A^ mice have been previously described [14], our study provides one of the initial pieces of evidence indicating similar alterations in cortical neurons of this murine model.

### 2.2. Acute Trimetazidine Treatments Protect Mitochondria in Primary Spinal and Cortical Cell Cultures

In a previous study by Scaricamazza et al. [33], we demonstrated that chronic administration of TMZ to SOD1^G93A^ mice restores proper mitochondrial metabolism in both spinal cord and skeletal muscle. To further investigate the effects of TMZ directly administered to PSC and PCC cultures, we analyzed the bioenergetic profiles following ON treatments with 10 μM TMZ. Our results indicate that TMZ treatment improved basal respiration in both primary cultures (Figure 2a,b), as well as maximal respiration, spare respiratory capacity and ATP-linked respiration (Figure 2c,d). Furthermore, our analysis on the contribution of glycolysis and oxidative metabolism to ATP production reveals that TMZ treatment increased ATP levels in both primary cell cultures, mainly through mitochondrial function rather than glycolysis (Figure 2e,f).

Moreover, we previously observed the failure of mitochondrial activity of Complex I and Complexes II/III in isolated mitochondria from spinal cord and skeletal muscles of SOD1^G93A^ mice, and we showed that chronic administration of TMZ restores these dysfunctions [16]. In this study, we observed that TMZ treatment has a comparable impact on both PSCs and PCCs, restoring the diminished activity of Complex I and Complexes II/III (Figure 3a,b).

Since we were unable to evaluate Complex IV activity using Seahorse technology, we performed a spectrophotometric analysis of the functional activity of Complex IV in both PSCs and PCCs and did not detect any specific damage in this complex, in accordance with Scaricamazza et al., 2020 (Appendix A) [14].

### 2.3. Ultrastructural Mitochondrial Morphology Is Preserved in Primary Spinal and Cortical Cell Cultures Administered with Trimetazidine Treatment

Ultrastructural analysis of PSCs and PCCs obtained from SOD1^G93A^ mice clearly showed alterations in mitochondrial morphology (Figure 4a,b). In both PSC and PCC cultures obtained from wild-type (WT) mice, mitochondria appeared to be elongated with a length-to-width ratio of approximately 4:1 and a length of about 2 µm. Mitochondria localization appeared to be perinuclear and exhibited regular and well-organized cristae in a parallel arrangement. In contrast, in SOD1^G93A^-derived PSCs and PCCs, mitochondria displayed significant alterations in the inner mitochondrial membrane (IMM) while maintaining apparent integrity of the outer mitochondrial membrane (OMM) (Figure 4a,b). Specifically, mitochondria in SOD1^G93A^ PCCs showed fewer, poorly developed and misaligned cristae (Figure 4b). SOD1^G93A^-derived PSCs exhibited slightly reduced density of mitochondrial cristae compared to the WT, but their appearance was severely deranged and fragmented (Figure 4a).

Interestingly, TMZ treatment led to improved mitochondria morphology in both SOD1^G93A^-mice-derived PSCs and PCCs, resulting in a marked reduction in aberrant mitochondrial frequencies (Figure 4c,d). Conversely, WT cell cultures did not show any noticeable changes of mitochondrial morphology after treatment (Figure 4c,d).

In accordance with the ultrastructural results, detected by both immunofluorescence and MiNa2.0 toolset analysis, SOD1^G93A^-mice-derived PSCs and PCCs had altered mitochondrial morphology (Figure 4e,f). In detail, in both SOD1^G93A^ primary cells, we observed a higher number of punctate-shaped mitochondria and a reduction in rod-like-shaped ones, revealing a clear increase in mitochondrial fragmentation (Figure 4g,i). These alterations were further supported by a significant reduction in mitochondrial footprint, mean branch length and branch count in the context of SOD1^G93A^ background PCCs and PSCs (Figure 4h,j), indicating a perturbation in the mitochondrial network. TMZ treatment led to a substantial improvement in the measured parameters related to mitochondrial morphology, and this positive effect was evident in both primary cell lines (Figure 4e–j).

### 2.4. Trimetazidine Treatment Stimulates Autophagy and Mitophagy Processes

Ultrastructural analyses revealed that TMZ treatment induces the activation of the autophagy process, increasing the number of autophagosomes in both PSCs and PCCs regardless of the genotype and cell type (Figure 5a,b). These double-membrane-limited vacuoles are predominantly localized in the perinuclear region and contain heterogeneous content, such as multilamellar bodies, partially degraded organelles and amorphous material. Notably, mitochondrial fragments are identifiable within several vesicles (Figure 5a,b).

Interestingly, in these primary cell cultures, we observed a moderate but statistically significant decrease in the autophagosome number in SOD1^G93A^-mice-derived PSCs and PCCs compared to in those from WT mice (Figure 5c,d).

To overcome limitations associated with primary cell cultures, we employed the murine motoneuronal NSC34 cell line stably expressing human SOD1 proteins that are either WT or mutated SOD1^G93A^. These cells displayed mitochondrial dysfunctions like those observed in PSCs and PCCs, and treatment with TMZ demonstrated its efficacy in restoring the altered parameters. Specifically, TMZ treatment restored basal respiration, ATP-linked respiration, maximal respiration and spare respiratory capacity (Appendix A). 

Subsequently, we employed these cells to further investigate the capacity of TMZ to induce autophagy/mitophagy processes. Immunofluorescence analysis of LC3 puncta revealed a significant increase in the number of autophagosomes upon TMZ treatment, particularly in SOD1^G93A^-expressing cells (Figure 5e,f). Moreover, under the same experimental conditions, the colocalization of LC3 with the mitochondrial marker ATBP increased, irrespective of the genotype, although it was more pronounced in SOD1^G93A^ -expressing cells, indicating the occurrence of mitochondria degradation via mitophagy (Figure 5f). To assess autophagy/mitophagy flux, cells were cultured in the presence and absence of the late-stage autophagy inhibitor tetra-ammonium chloride (NH_4_Cl), and the protein expression of the LC3II-lipidated form and phospho-ubiquitin (S65) levels were evaluated. Both markers increased with TMZ treatment, supporting our findings regarding the role of TMZ in promoting mitophagy (Figure 5g,h).

### 2.5. Trimetazidine Treatment Ameliorates Mitochondrial Functionality of ALS-Patient-Derived PBMCs

Finally, we conducted an analysis of bioenergetic parameters in PBMCs obtained from both ALS patients and control subjects. While no significant differences were observed in basal respiration and ATP-linked respiration between ALS patients’ PBMCs and those of healthy subjects, TMZ treatment resulted in a notable increase in both parameters (Figure 6a,b). Interestingly, the OCR related to the maximal respiration of ALS patients’ PBMCs was lower compared to that of healthy control subjects (Figure 6c). Functionally, this reduction suggests potential alterations in the electron transport chain, which become detectable when the mitochondria are challenged. Remarkably, TMZ treatment restored the maximal respiration to a level comparable to that of the control PBMCs (Figure 6c). Additionally, we observed that TMZ increased the value of the spare respiratory capacity, irrespective of the ALS background (Figure 6d).

Furthermore, correlation analysis between main clinical features (namely age, BMI, disease duration, ALS-FRS-R score) and mitochondrial bioenergetic profile showed no significant correlation.

## 3. Discussion 

TMZ is a well-established medication that has been used in Europe for over 40 years for the treatment of angina pectoris. TMZ directly improves myocardial metabolism by modulating fatty acid beta oxidation rather than indirectly affecting hemodynamics. Numerous studies have confirmed its positive effects on myocardial fibrosis, apoptosis and inflammation [22,37,38,39].

TMZ is often defined a “metabolic modulator” due to its ability to improve energy metabolism in heart failure; it reduces fatty acid metabolism through the inhibition of the enzyme ACAA2 and by enhancing glucose metabolism through the activation of the rate-limiting enzyme Pyruvate Dehydrogenase (PDH), involved in glucose aerobic oxidation [24,40]. Nevertheless, the effects of TMZ on cardiac muscle energy metabolism have already been questioned, suggesting that the capability of the drug to enhance the ATP production could be attributable to other mechanisms [41].

In this context, further studies described the ability of TMZ to enhance cardiac function by modulating autophagy in cardiomyocytes [42] and to counteract inflammation by significantly reducing the levels of inflammatory markers such as IL-1β, IL-6 and TNF-α in the bloodstream [43,44].

In addition to its effects on cardiac muscle, several studies highlighted the role of TMZ in improving energy metabolism in skeletal muscle of ageing and cancer-related cachexia murine models. In these models, chronic treatment with TMZ did not impact skeletal muscle mass but significantly increased muscle strength [31,45].

Interestingly, the World Anti-Doping Agency (WADA) has recently classified this drug as a prohibited substance due to its impact on skeletal muscle metabolism [45]. Indeed, TMZ, in different murine experimental paradigms, elicits effects akin to those achieved through physical exercise, including enhanced grip strength, a transition towards a slow myofiber phenotype, decreased blood glucose levels, up-regulation of PGC1α (a regulator of oxidative metabolism and mitochondrial biogenesis) and partial restoration of the altered cross-sectional area of myofibers [31,45]. Notably, TMZ stimulates the expression of the slow myosin heavy chain isoform and facilitates the growth of small-sized myofibers [31,45].

In this context, in a recent study, we explored the effects of TMZ in a murine model of ALS, demonstrating that chronic treatment significantly improves the pathological phenotype of the disease. The study not only highlighted the positive effects on muscle metabolism but also revealed that the drug affects the entire motor unit, thereby preserving the stability of the NMJ, inhibiting axonal dying back and preventing the loss of spinal cord motor neurons. This study has therefore shed light on the neuroprotective function of TMZ, which has been poorly explored thus far. Most previous studies, indeed, have primarily focused on the role of TMZ in skeletal and cardiac muscle, while few papers highlighted the diverse effects of TMZ on the nervous system, including its neuroprotective properties, modulation of mitochondrial function and potential therapeutic applications in neurological conditions such as epilepsy, glaucoma and nerve injuries [46,47,48,49,50,51].

Based on our groundbreaking findings regarding the effects of TMZ in a neurodegenerative disorder, we used neuronal cell models to investigate the drug’s mechanism of action within the context of ALS.

First, we highlighted that both PSC and PCC cultures, obtained from SOD1^G93A^ mice, had altered mitochondrial functionality. ATP-linked respiration, maximal respiration and spare respiratory capacity, the latter being a parameter that specifically outlines the ability of mitochondria to adapt energy production to the request, appeared affected in both SOD1^G93A^-derived cell cultures. This outcome was partially expected in the primary cell cultures derived from the spinal cord as it is the CNS area primarily affected in SOD1^G93A^ mice and more generally in ALS. However, it is particularly interesting to have identified identical alterations in the motor cortex cultures as well. This finding, in fact, represents one of the few pieces of evidence indicating energy metabolism alterations in neuronal cell cultures obtained from a distinct CNS region compared to the spinal cord in this animal model [52].

These mitochondrial alterations, however, did not affect survival of either PSC or PCC cultures obtained from SOD1^G93A^ mice, which, instead, become more sensitive to specific stressors such as kainate or nitric oxide compared to the same cultures obtained from WT mice [35,53].

Notably, in the same PCC cultures, by means of Patch clamp techniques, the authors demonstrated alterations in membrane repolarization and intrinsic hyperexcitability. These changes were attributed to a higher current density of the persistent sodium current, specifically in the neurons obtained from transgenic SOD1^G93A^ mice [54,55].

Note that the impairment in energy metabolism could partially justify these described alterations in membrane electrophysiology, as suggested by the computational paper from Le Masson and colleagues [56].

Moreover, the treatment with TMZ successfully restored the altered mitochondrial morphology observed in PSC and PCC cultures of SOD1^G93A^ mice, as revealed by electron microscopy analysis. Interestingly, in TMZ-treated cell cultures, the analysis also revealed a notable increase in autophagosomes, which were identified as lysosomes containing remnants of mitochondria. This observation suggests an ongoing process of mitophagy. Further confirmation of this suggestion came from the analysis of autophagic/mitophagic flux in the same primary cell cultures and in murine motor neuron cell lines expressing the mutant SOD1^G93A^.

The capacity of TMZ to induce autophagy has been reported in several studies, primarily in cardiac and skeletal muscle. These investigations have demonstrated that TMZ can initiate the autophagy process in various mouse models of stress and heart failure [22,23,42,57,58].

The findings we obtained provide the initial evidence suggesting a direct involvement of TMZ in enhancing mitochondrial quality control in neuronal cell models of ALS, through the activation of autophagy/mitophagy flux. Dysregulation of the autophagy process has been extensively observed in animal and cellular models of ALS [59]. In fact, the evidence supporting impaired mitophagy in ALS has emerged from studies identifying mutations in genes such as OPTN and TBK1, which play crucial roles in regulating mitochondrial turnover [60,61,62].

It has been suggested that an inefficient quality control mechanism of mitochondria may contribute to the accumulation of dysfunctional mitochondria observed in ALS motor neurons [59,63,64]. 

In this context, a recent study revealed an impairment in both mitochondrial functionality and the mitophagy process in the spinal cord of symptomatic SOD1^G93A^ mice. This dysfunction was found to be associated with the overexpression of Translocator Protein 18kD (TSPO), a mitochondrial factor known to be involved in the regulation of mitophagy [65]. 

Finally, in a different experimental setting, researchers reported the presence of morphologically altered mitochondria in peripheral blood mononuclear cells (PBMCs) of sporadic ALS (sALS) patients associated with an inefficient turnover of damaged mitochondria [66]. In our study, we examined PBMCs from a panel of ALS patients, including mostly sporadic cases and two familial cases, compared to control PBMCs and found that treatment with TMZ restored the observed alterations. This further suggests the potential role of TMZ in inducing autophagy/mitophagy flux.

In conclusion, our findings confirm the significance of targeting mitochondrial dysfunction as a therapeutic strategy to counteract the ALS phenotype, in line with the extant preclinical literature [67]. 

Furthermore, we propose a mechanism by which TMZ enhances mitochondrial performance in motor neuron cells, although the detailed molecular pathway through which TMZ activates autophagy requires further investigation. This understanding is crucial for developing TMZ derivatives with enhanced therapeutic efficacy and greater target specificity within the context of ALS.

## 4. Materials and Methods

### 4.1. Antibodies 

The antibodies used in this study are summarized in Table 1.

### 4.2. Animals

All the animal procedures were performed following the European guidelines for the use of animals in research (2010/63/EU) and the requirements of Italian laws (D.L. 26/2014). All the procedures were approved by the Italian Ministry of Health (protocol number 293/2021-PR). Transgenic hemizygous SOD1^G93A^ male mice (B6.Cg-Tg [SOD1 G93A]1Gur/J) were obtained from The Jackson Laboratory (Bar Harbor, ME, USA; RRID:MGI:4835776) and then crossbred with C57BL/6 females. The embryos were genotyped by PCR using hSOD1 (forward 5′-CATCAGCCCTAATCCATCTGA-3′; reverse 5′-CGCGACTAACAATCAAAGTGA-3′) oligos with Interleukin-2 (IL-2) (forward 5′-TAGGCCACAGAATTGAAAGATCT-3′; reverse 5′-GTAGGTGGAAATTCTAGCATCATCC-3′) as housekeeping gene.

### 4.3. Primary Cell Cultures and Cell Lines

To obtain PSC and PCC cultures, 15-day-old pregnant C57BL/6J females were anesthetized using Rompum (xylazine, 20 mg/mL/L 0.5 mL/kg/L, Bayer, Milan, Italy) with Zoletil (tiletamine and zolazepam, 100 mg/mL/L, 0.5 mL/kg/L, Virbac, Milan, Italy), as previously described [35,36]. Each neural tube was dissected, and cortices were removed, singularly incubated for 10 min in 0.025% trypsin, dissociated by gentle trituration and plated. The resulting mixed cultures were seeded on a poly-D-lysine/Laminin-coated Seahorse plate or glass cover slips and maintained in Neurobasal medium supplemented with B-27, 1 mM glutamine, 100 IU/mL penicillin/100 μg/mL streptomycin and 5% fetal bovine serum and, for spinal cord, also 5% horse serum. Cultures were maintained in a 37 °C humidified incubator in 5% CO_2_ atmospheres. Three days after plating, the medium was replaced with Neurobasal supplemented with B-27 for spinal cord cultures and with Neurobasal supplemented with B-27 and 0.3 mM L-glutamine for cortical cultures. SOD1^G93A^ expression was confirmed in both PSCs and PCCs by WB analysis (Appendix A). Moreover, to suppress glia proliferation, the Ara-C compound was added to a final concentration of 10 μM, and GFAP and CX43 expression was evaluated (Appendix A). Experiments on cultures were performed between days 13 and 15 in vitro, unless otherwise indicated. Unused embryo tissues were subjected to PCR to assess the expression of transgenic human SOD1^G93A^ gene, and, from this analysis, about 40 embryos were identified as transgenic (SOD1^G93A^) and 40 as WT.

Mouse motoneuronal cell line NSC34 was stably transfected with the pTet-ON plasmid (Clontech, Palo Alto, CA, USA) to produce inducible cell lines expressing human WT SOD1 and SOD1^G93A^ following 48 h exposure to Doxycycline according to the procedure described by Ferri and colleagues [53].

In accordance with Belli et al. [32], PSC and PCC cultures were treated ON with 10 μM of TMZ dissolved in culture medium before the experiments.

NSC34 cell lines were untreated or treated for 48 h with 10 mM NH_4_Cl and ON with 10 μM TMZ, both dissolved in culture medium, prior to being analyzed.

### 4.4. Amyotrophic Lateral Sclerosis Patient Biosampling and PBMC Preparation

Fourteen ALS patients were enrolled at Tor Vergata University Hospital (Rome, Italy) in 2022. ALS was diagnosed according to the revised El Escorial criteria [68]. Subjects with main acute/chronic infectious/inflammatory/internal diseases and/or abnormal blood cell count were excluded.

For each patient, demographics, anthropometrics and medical history were collected. Patients were clinically assessed with ALS-FRS-R and King’s score. Genetic testing for SOD1, TARDBP, FUS and C9orf72 was systematically performed in all ALS patients.

Table 2 summarizes demographic and clinical data of the study population.

As previously reported [69], all participants underwent venous blood sampling (20 mL) in the morning, after overnight fasting (morning drugs allowed). Blood was immediately processed to separate PBMCs through density gradient centrifugation with Ficoll-Hypaque (GE Healthcare Life Sciences, Little Chalfont, UK) according to standard procedures [69]. PBMCs were carefully frozen in cryoSFM medium and stored in liquid nitrogen. Once thawed, PBMCs were suspended in complete RPMI before cell treatments, then cells were treated ON with 10 µM TMZ and subsequently underwent bioenergetics analyses using a Seahorse XF96e Analyzer (Seahorse Bioscience-Agilent, Santa Clara, CA, USA).

The study was approved by the local ethics committee following the principles of the Declaration of Helsinki. All participants signed to indicate informed consent.

### 4.5. Electrophoresis and Western Blotting

Protein obtained from TMZ-treated and untreated PSC and PCC cultures, and NSC34 cell lines, was separated by SDS–polyacrylamide gel electrophoresis and transferred onto nitrocellulose membranes (Perkin Elmer, (Hopkinton, MA, USA) Cat# NBA085B). Membranes were blocked for 1 h in Tris-buffered saline solution with 0.1% Tween-20 (TBS-T) containing 5% non-fat dry milk and incubated for 2 h at room temperature with the indicated primary antibodies diluted in TBS-T containing 2% non-fat dry milk. Primary antibodies were detected using the appropriate peroxidase-conjugated secondary antibody diluted in TBS-T containing 1% non-fat dry milk, then washed and developed using enhanced chemiluminescence (BIO-RAD Clarity^TM^ Western ECL substrate Cat# 170–5061).

Densitometric analyses were performed using the ImageJ software 1.50i program (US National Institutes of Health, Bethesda, MD, USA, https://imagej.nih.gov/ij/, 1997–2016). The apparent molecular weight of proteins was determined by calibrating the blots with pre-stained molecular weight markers (Bio-Rad Laboratories, Cat# 161-0394).

### 4.6. Immunofluorescence Analysis

For immunofluorescence analysis, control and TMZ-treated PSC and PCC cultures, and NSC34 cell lines, were fixed with 4% paraformaldehyde in 0.1 M phosphate buffer pH 7.4 and subsequently permeabilized with 0.1% Triton X-100 for 8 min and washed in phosphate-buffered saline (PBS). Samples were saturated for 30 min with 2% horse serum in PBS and incubated with primary antibodies, followed by the appropriate secondary antibody. After rinsing in PBS, the cells were counterstained with DAPI in phosphate-buffered solution (1 microg/mL, SIGMA, (Darmstadt, Germany)). Immunofluorescence was examined under confocal laser scanning microscope (Zeiss Airyscan LSM800, (Ostfildern, Germany)) and images analyzed by Adobe Photoshop 2020, and ImageJ (US National Institutes of Health, Bethesda, MD, USA, https://imagej.nih.gov/ij/, 1997–2016) software.

### 4.7. Bioenergetic Analysis

Mitochondrial functionality was determined using a Seahorse XF96e Analyzer (Seahorse Bioscience—Agilent, Santa Clara, CA, USA).

The 96-well Seahorse plate was coated with Cell-Tak (Corning (Tewksbury, MA, USA)and TMZ-treated and untreated cells were seeded as follows: PSCs and PCCs, 10.000 cells per well, maintained for 14 days as described above; NSC34 cell lines, 7.000 cells per well; PBMCs, 250.000 cells per well.

The assays were performed according to Agilent’s recommendations. Briefly, in the Cell Mito Stress Test and ATP Real-Time Rate Assay, growth medium was replaced with XF test medium (Eagle’s modified Dulbecco’s medium, 0 mM glucose, Agilent Seahorse (Santa Clara, CA, USA)) supplemented with 1 mM pyruvate, 10 mM glucose and 2 mM L-glutamine, with pH adjusted to 7.4. In the Electron Flow Assay, growth medium was replaced with Respiration Buffer (250 mM Sucrose, 15 mM KCl, 1 mM EGTA, 5 mM MgCl_2_, 30 mM K_2_HPO_4_) supplemented with pyruvate (10 mM), malate (2 mM), ADP (4 mM) and XF PMP reagent (1 nM).

Before the assay, the cells were incubated in a 37 °C incubator without CO_2_ for 45 min to allow them to pre-equilibrate with the assay medium.

The Cell Mito Stress Test was performed by measuring first the baseline oxygen consumption rate (OCR), followed by sequential OCR measurements after injection of oligomycin (1.5 µM), carbonyl cyanide 4-(trifluoromethoxy) phenylhydrazone (1 µM) and rotenone (0.5 µM) + antimycin A (0.5 µM). In the ATP Real-Time Rate Assay, measurements were taken after injection of oligomycin (1.5 µM) and rotenone (0.5 µM) + antimycin A (0.5 µM). In the Electron Flow Assay, the XFe96 cartridge was loaded with drugs at a final concentration of rotenone 2 μM, succinate 10 mM, antimycin A 4 μM, ascorbate and TMPD 10 mM and 100 μM.

All data were analyzed using XFe Wave software 2.6 and presented as point-to-point OCR normalized to protein content (pmol/minute/μg-protein). In detail, each individual OCR measurement was normalized based on the protein content of its respective well, measured after completion by standard colorimetric assay.

### 4.8. Spectrophotometric Assays

The activity of respiratory chain Complex IV was evaluated according to Salvatori et al. [70]. Briefly, PSCs and PCCs were sonicated (UP200S Ultrasound Technology, Hielscher (Teltow, Germany), 20% amplitude, 0.5 cycles for 30 s), and Complex IV activity was assayed following the oxidation of ferricytochrome c at 540 nm in a 1 mL cuvette at 30 °C. Spectrophotometric determination of citrate synthase activity was measured, and values were used as standard internal controls.

### 4.9. Electron Microscopic Analysis

For ultrastructural analysis, PSC and PCC cultures were plated in chamber slides (Lab-Tek™ II Chamber Slide System, ThermoFischer Scientific, (Waltham, MA, USA)) and fixed in a mixture of 2% formaldehyde and 1% glutaraldehyde in 0.1 M cacodylate buffer, pH 7.4, for 45 min at 4 °C. After washing, samples were post-fixed in 1% osmium tetroxide for 45 min in the dark, then rinsed in distilled water and finally stained en bloc with UranyLess (Electron Microscopy Science, Foster City, CA, USA) for 1 h in the dark. Samples were gradually dehydrated in ethanol, infiltrated with a mixture of ethanol and epoxy embedding medium (Sigma-Aldrich, Cat# 45359-1EA-F, Burlington, MA, USA) 1:1 for 90 min, then allowed to polymerize in absolute resin at 60 °C for 72 h. Embedded cells were mounted on aluminum stubs by means of a bi-adhesive carbon disk made conductive by depositing a thin layer of gold with a K550 sputter coater (Emitech, Ashford, UK) and analyzed with a Helios Nanolab 600 Dual Beam (FIB/SEM) (FEI Company, Hillsboro, OR, USA) at the LIME facility, University Roma Tre. Samples were milled at regions of interest using the FIB column operated with a voltage of 30 KV and a current of 6.5 nA, and micrographs were acquired by using the SEM column, detecting backscattered electrons with an accelerating voltage of 2 kV and an applied current of 0.17 nA.

### 4.10. Mitochondria Morphological Analysis

Mitochondria morphology was analyzed using the Mitochondria Network Analysis tool (MiNA), available at https://github.com/stuartlab/MiNA (accessed on 6 March 2024), run on the ImageJ interface. The analysis was performed following the instructions of Valente et al. [71]. Cells were seeded into coverslips and immune-stained with anti-TOMM20; antibody images were obtained using Confocal Zeiss LSM800 laser-scanning microscope (Ostfildern, Germany).

Briefly, images were imported into ImageJ and processed to 8-bit images in grayscale. To enhance the quality, the images were pre-processed with local contrast CLAHE. The images were then binarized to generate black foreground mitochondria images over a white background, then converted to a skeleton. The skeleton was further analyzed via the ImageJ plugin “analyze skeleton”, which measures the length of each branch, the number of branches in each skeleton and how spatially related the branches are. This outcome was further processed through the MiNA macro plugin to generate parameters describing mitochondrial network morphology (e.g., mitochondria footprint, branch length, etc.). Ten cells per sample were analyzed to obtain measurements for mitochondria footprint, mitochondria branch length mean and mitochondria network branching mean.

### 4.11. Statistical Analysis

Data were analyzed using Kaplan–Meier test, Student’s *t*-test (two-group comparison) and one-way or two-way ANOVA. Post hoc analysis was carried out using Bonferroni or Tukey tests. Correlations were evaluated using Spearman’s test. Statistical significance was set at *p* < 0.05. Data are presented as mean ± SEM. Statistical analysis was carried out using GraphPad Prism 5 software.

## Figures and Tables

**Figure 1 ijms-25-03251-f001:**
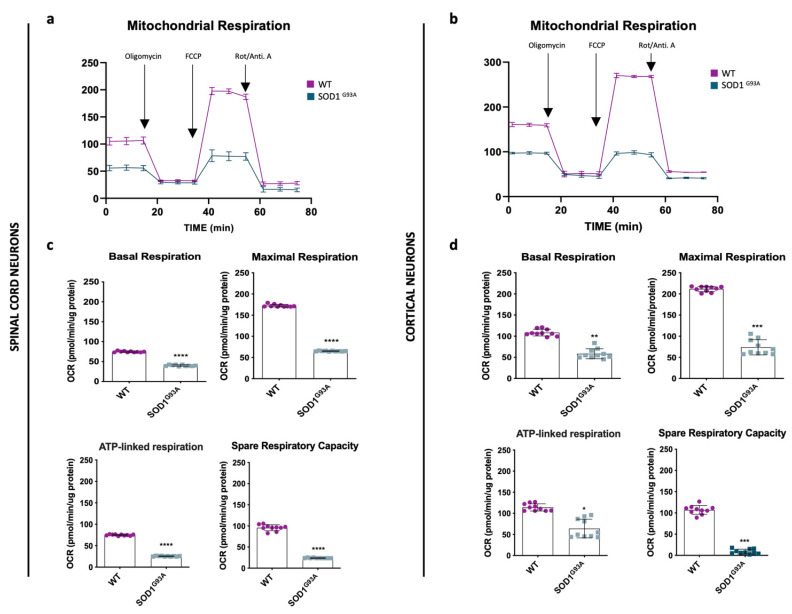
Mitochondrial functionality is impaired in primary spinal cord and cortical cell cultures obtained from SOD1^G93A^ mice. (**a**,**b**) Representative profile of oxygen consumption rate (OCR) obtained through Cell Mito Stress Test from primary spinal cord (**a**) and cortical (**b**) cell cultures obtained from WT and SOD1^G93A^ mice. OCR was measured after the addition of drugs: oligomycin, FCCP, rotenone and antimycin A (see arrows). The time on the *x*-axis represents the time point when each measurement was performed. (**c**,**d**) The histograms show the individual parameters of basal respiration, ATP-linked respiration, maximal respiration and spare respiratory capacity obtained for primary spinal cord (**c**) and cortical (**d**) cell cultures. All data were analyzed with XFe Wave software and are expressed as OCR pmol/minute/μg-protein. Data are presented as means ± SEM, * *p* < 0.05, ** *p* < 0.01, *** *p* < 0.001, **** *p* < 0.0001 compared with wild type, n ≥ 10 per group, each (n) with ≥ 12 technical replicates. *p* values were obtained using unpaired Student’s *t*-test.

**Figure 2 ijms-25-03251-f002:**
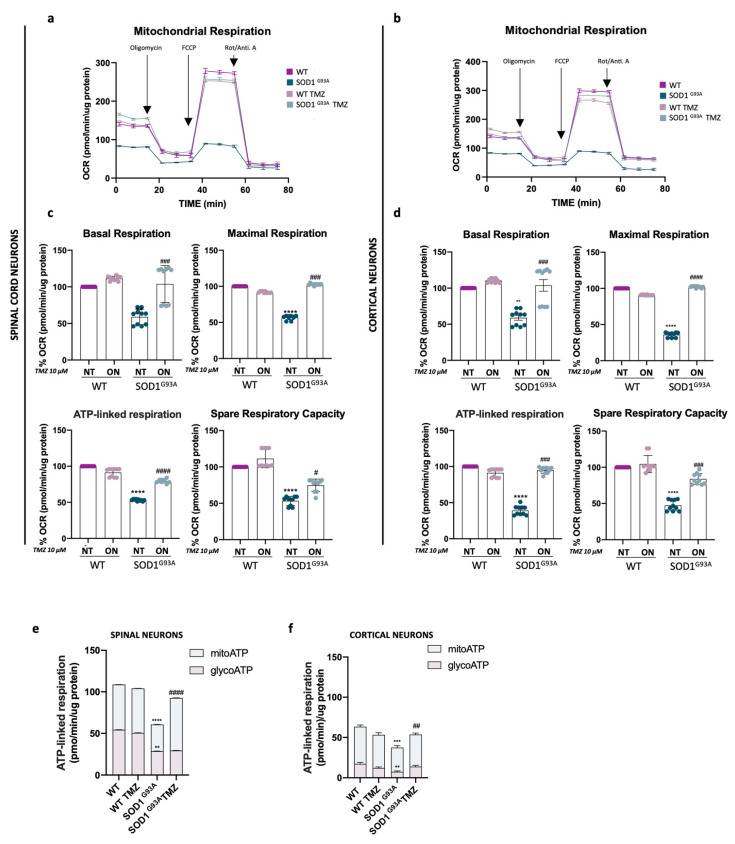
Acute Trimetazidine treatments protect mitochondria in primary spinal cord and cortical cell cultures. (**a**,**b**) Representative profile of oxygen consumption rate (OCR) obtained through Cell Mito Stress Test from primary spinal cord (**a**) and cortical (**b**) cell cultures obtained from WT and SOD1^G93A^ mice untreated or treated overnight (ON) with 10 μM of Trimetazidine (TMZ). (**c**,**d**) The histograms show the data obtained from the analysis of the Cell Mito Stress Test performed on primary spinal cord (**c**) and cortical cultures (**d**) obtained from WT and SOD1^G93A^ mice. Mitochondrial respiration expressed as normalized OCR after the addition of oligomycin, FCCP, rotenone and antimycin. The value of 100% was arbitrarily assigned to values obtained from NT WT. (**e**,**f**) Quantification of total, glycolytic and mitochondrial ATP-linked respiration obtained by Seahorse XF real-time ATP rate assay in primary spinal cord (**e**) and cortical cultures (**f**) following ON treatment with 10 µM TMZ. All data were analyzed with XFe Wave software and are expressed as OCR pmol/minute/μg-protein. Data are presented as means ± SEM, ** *p* < 0.01, *** *p* < 0.001, **** *p* < 0.0001 compared with wild type; # *p* < 0.05, ## *p* < 0.01, ### *p* < 0.001, #### *p* < 0.001 compared with untreated SOD1^G93A^, n ≥ 10 per group, each (n) with ≥ 6 technical replicates. *p* values were obtained using parametric one-way ANOVA with Bonferroni post hoc test.

**Figure 3 ijms-25-03251-f003:**
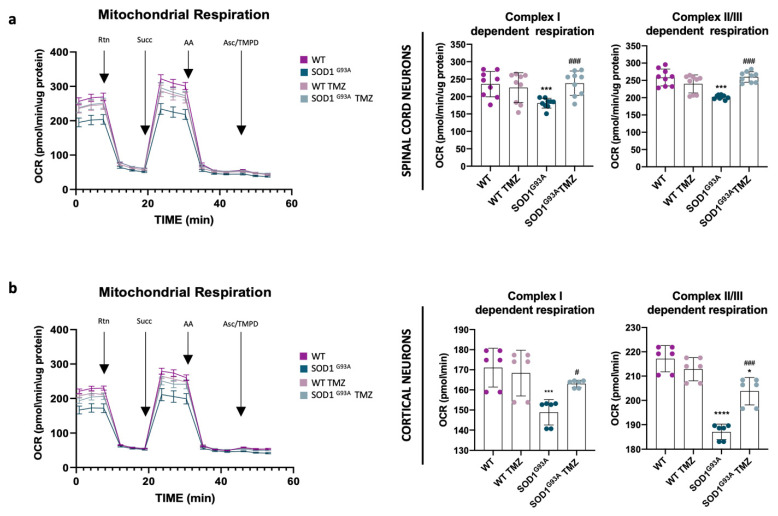
Acute Trimetazidine treatments restore mitochondria complex function in primary spinal cord and cortical cell cultures. (**a**,**b**) In the left panels, representative profiles of oxygen consumption rate (OCR) obtained through Electron Flow Assay from primary spinal cord (**a**) and cortical (**b**) cell cultures obtained from WT and SOD1^G93A^ mice untreated or treated overnight (ON) with 10 μM of Trimetazidine (TMZ) are displayed. In the right panels, quantification of respiration (OCR) dependent on the activity of Complex I and Complexes II/III in the presence of rotenone, succinate and antimycin A is displayed, respectively, in primary spinal cord (**a**) and cortical cell cultures (**b**) obtained from WT and SOD1^G93A^ mice untreated or ON treated with 10 μM TMZ. All data were analyzed with XFe Wave software and are expressed as OCR pmol/minute/μg-protein. Data are presented as means ± SEM, * *p* < 0.05, *** *p* < 0.001, **** *p* < 0.0001 compared with wild type; # *p* < 0.05, ### *p* < 0.001 compared with untreated SOD1^G93A^, n ≥ 6 per group, each (n) with ≥6 technical replicates. *p* values were obtained using parametric one-way ANOVA with Bonferroni post hoc test.

**Figure 4 ijms-25-03251-f004:**
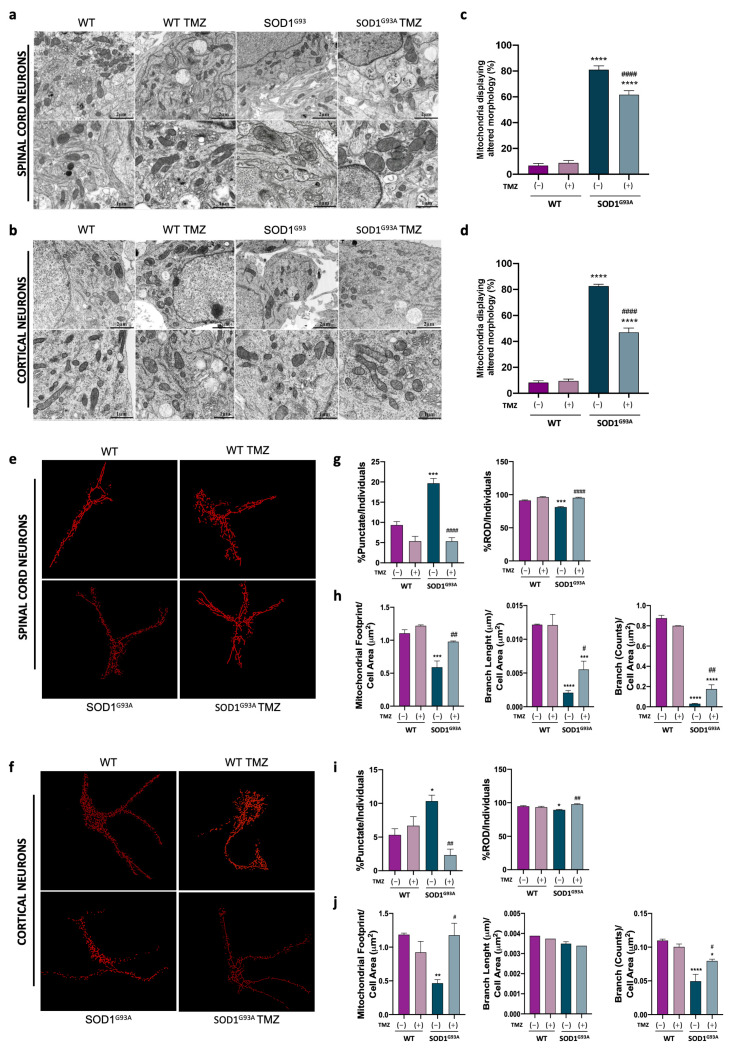
Ultrastructural mitochondrial morphology is preserved in primary cell cultures administered with Trimetazidine treatment. (**a**,**b**) Representative FIB/SEM micrographs illustrating mitochondrial morphology in primary spinal cord (**a**) and cortical (**b**) cell cultures obtained from WT and SOD1^G93A^ mice untreated or ON treated with 10 μM of Trimetazidine (TMZ) at different scales (scale bars: 1 μm and 2 μm) and (**c**,**d**) relative quantification of mitochondria displaying altered morphology. (**e**,**f**) Representative skeletonized mitochondria images, obtained from primary spinal cord (**e**) and cortical (**f**) cell cultures from WT and SOD1^G93A^ mice untreated or ON treated with 10 μM TMZ. Cell cultures were previously immunolabeled with the mitochondrial marker ATPB and then analyzed through the Mitochondria Network Analysis tool (MiNA). (**g**,**i**) Analysis of the mitochondrial morphology expressed as percentages of punctate-shaped (left panels) and rod-like-shaped (right panels) mitochondria of primary spinal cord (**g**) and cortical (**i**) cell cultures obtained from WT and SOD1^G93A^ mice untreated or ON treated with 10 μM TMZ. (**h**,**j**) Analysis of the mitochondrial morphology as in (**g**,**i**) expressed as mitochondria footprint (left panels), mean branch length (middle panels) and count branches (right panels). All values were normalized for the cell area (μm^2^). Data are presented as means ± SEM, * *p* < 0.05, ** *p* < 0.01, *** *p* < 0.001, **** *p* < 0.0001 compared with wild type; # *p* < 0.05, ## *p* < 0.01, #### *p* < 0.001 compared with untreated SOD1^G93A^, n ≥ 4 per group and n = 35 cells each for the MiNA. *p* values were obtained using parametric one-way ANOVA with Bonferroni post hoc test.

**Figure 5 ijms-25-03251-f005:**
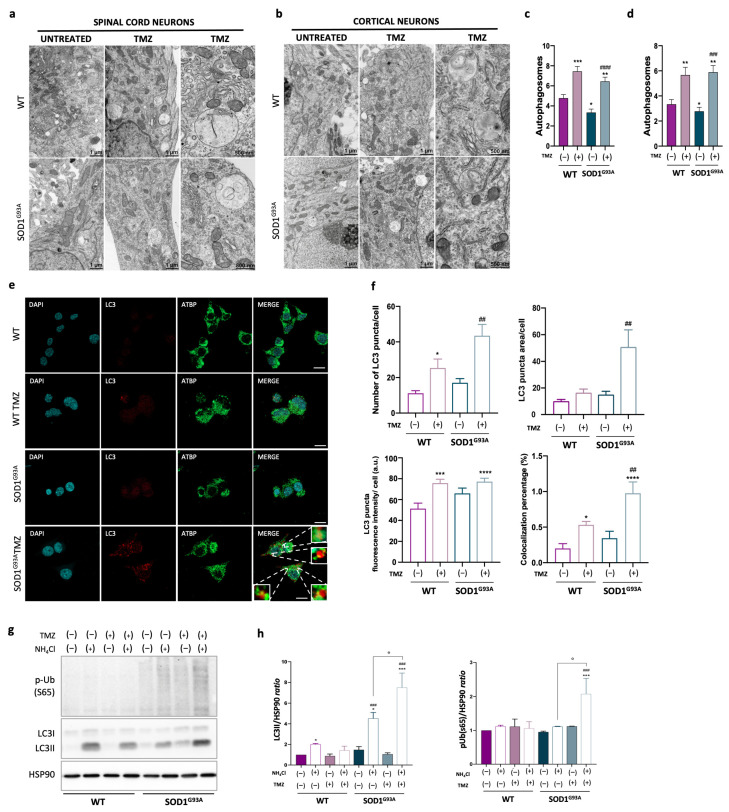
Trimetazidine treatment stimulates autophagy and mitophagy processes. (**a**,**b**) Representative FIB/SEM micrographs illustrate that Trimetazidine (TMZ) treatment in primary spinal cord (**a**) and cortical (**b**) cell cultures stimulates the formation of autophagosomes. (**c**,**d**) Relative quantification of data obtained in (**a**,**b**). (**e**) Representative confocal immunofluorescence images showing LC3 and ATBP staining and LC3/ATBP colocalization (merge) in NSC34 cell lines treated or not with 10 μM of TMZ ON as indicated. Scale bar 10 μm (**f**) Quantification of the LC3 puncta average number, area, fluorescence intensity and LC3/ATBP colocalization percentage from the images as in (**c**) performed using ImageJ quantification tool. (**g**) Representative Western blot image of p-Ub(S65), LC3I and its lipidated form LC3II, obtained in protein extracts from NSC34 cell lines treated or not with TMZ and NH_4_Cl as indicated; HSP90 was used as loading control. (**h**) Densitometric analysis of LC3II/HSP90 and p-Ub(S65)/HSP90 ratios from n = 4 independent experiments. Data are presented as means ± SEM, * *p* < 0.05, ** *p* < 0.01, *** *p* < 0.001, **** *p* < 0.0001 compared with wild type; ## *p* < 0.01, ### *p* < 0.001, #### *p* < 0.001 compared with untreated SOD1^G93A^; ° *p* < 0.05 compared with untreated NH_4_Cl. For ultrastructural analysis, n was ≥ 4 per group with at least 20 fields each. For immunofluorescence analysis, n was at least ≥ 4 per group with at least 35 cells. Values were obtained using parametric one-way ANOVA with Bonferroni post hoc test.

**Figure 6 ijms-25-03251-f006:**
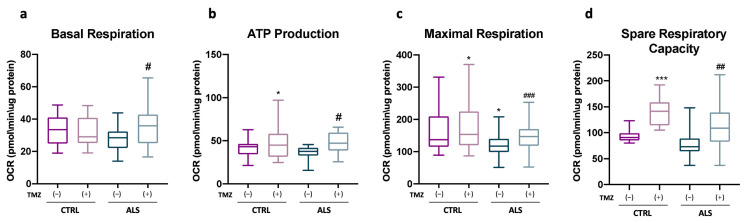
Trimetazidine treatment improves mitochondrial performances of PBMCs from ALS patients. Measurement of the rate of oxygen consumption ratio (OCR) in PBMCs isolated from ALS patients and healthy subjects treated or untreated with 10 μM of Trimetazidine (TMZ) as indicated. The histograms show the data obtained from the analysis of the Cell Mito Stress Test. In detail: (**a**) basal respiration, (**b**) ATP-linked respiration, (**c**) maximal respiration and (**d**) spare respiratory capacity. All data were analyzed with XFe Wave software and are expressed as OCR pmol/minute/μg-protein. Data are presented as means ± SEM, * *p* < 0.05, *** *p* < 0.001 compared with healthy subjects; # *p* < 0.05, ## *p* < 0.01, ### *p* < 0.001 compared with untreated healthy subjects; ALS patients, n = 14, healthy subjects, n = 20, each (n) with ≥ 6 technical replicates. *p* values were obtained using parametric one-way ANOVA with Bonferroni post hoc test.

**Table 1 ijms-25-03251-t001:** Primary and secondary antibodies used for Western blot (WB) and immunofluorescence (IF) analysis.

	Antibodies	Sources	Product	Dilution
Primary	Cu/Zn SOD	Rabbit polyclonal	Enzo (Lausen, Switzerland) (Cat#ADI-SOD-100D))	WB 1:1000
	β-actin	Mouse monoclonal	Sigma-Aldrich (Darmstadt, Germany) (Cat #MA1-140)	WB 1:1000
	TOM20	Rabbit polyclonal	Santa Cruz Biotechnology (Dallas, TX, USA) (Cat# sc-11415)	WB 1:1000
	VDAC	Mouse monoclonal	Santa Cruz Biotechnology (Dallas, TX, USA) (Cat# sc-8829)	WB 1:1000
	SMI-32	Mouse monoclonal	BioLegend (San Diego, CA, USA) (formerly Covance Antibody Products) (Cat# SMI-32P)	IF 1:1000
	Map2	Mouse monoclonal	Invitrogen (Carlsbad, CA, USA) (Cat# MA5-12826)	IF 1:500
	Connexin 43	Rabbit monoclonal	Sigma-Aldrich (Darmstadt, Germany) (Cat#ZRB1179-25UL)	WB 1:1000
	GFAP	Rabbit polyclonal	Invitrogen (Carlsbad, CA, USA) (Cat# PA5-16291)	IF 1:1000
	LC3B	Rabbit polyclonal	Novus Biologicals (Cambridge, UK) (Cat# NB100-2220)	WB 1:1000
	phospho-Ubiquitin (Ser65)	Rabbit monoclonal	Cell Signaling Technologies (Danvers, MA, USA) (Cat# BK62802)	WB 1:1000
	HSP90	Mouse monoclonal	Santa Cruz Biotechnologies (Dallas, TX, USA) (Cat# SC13119)	WB 1:1000
Secondary	HRP Conjugate	Goat anti-rabbit	Bio-Rad Laboratories (Richmond, CA, USA) (Cat#1706515; RRID: AB_2617112)	WB 1:2000
	HRP Conjugate	Goat anti-mouse	Bio-Rad Laboratories (Richmond, CA, USA) (Cat#170-6516, RRID: AB_11125547)	WB 1:2000
	Alexa Fluor 488	Donkey anti-mouse	Invitrogen (Carlsbad, CA, USA) (Cat# A11017)	IF 1:300
	Cy™3	Donkey anti-rabbit	Jackson Immuno-Research Laboratories (Farminton, CA, USA) (Cat# 65119)	IF 1:500

**Table 2 ijms-25-03251-t002:** Clinical data of ALS patients.

Sample	Gender	Age	Disease Duration (Months)	BMI	Clinical Phenotype	ALS-FRS-R	King’s Stage
sALS1	F	82	22	18	Spinal	39	2
sALS2	M	68	23	28	Spinal	34	3
sALS3	F	70	7	22.1	Spinal	32	3
sALS4	F	74	73	21.4	Spinal	13	4a
TARDBP	M	39	5	19.6	Spinal	44	2
sALS6	F	70	7	22.1	Spinal	32	3
sALS7	F	73	35	27.6	Spinal	37	2
sALS8	M	66	36	26.3	Spinal	46	2
sALS9	M	75	24	32	Spinal	37	3
sALS10	F	76	24	25.6	Spinal	42	1
sALS11	F	78	18	24.5	Bulbar	33	2
C9Orf72	F	58	19	21.9	Spinal	32	3
sALS13	M	69	13	23.7	Spinal	42	2
sALS14	M	55	12	-	Spinal	12	4a

## Data Availability

The authors confirm that the data supporting the findings of this study are available within the article and/or in its Appendix A.

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
