# Peer review of "Trimetazidine Improves Mitochondrial Dysfunction in SOD1G93A Cellular Models of Amyotrophic Lateral Sclerosis through Autophagy Activation"

_ijms, 2024, doi:10.3390/ijms25063251_

Round 1

Reviewer 1 Report

Comments and Suggestions for Authors

The manuscript by I. Salvatori and co-authors aims to study the effect of the metabolic modulator trimetazidine on the progression of mitochondrial dysfunction in cellular models of amyotrophic lateral sclerosis (primary cell cultures obtained from SOD1G93A mice and mouse motoneuronal cell lines NSC34) and sALS patient-derived peripheral blood mononuclear cells. The work showed that trimetazidine can improve mitochondrial respiration parameters, ATP production, electron transport chain complex activities, and mitochondrial morphology in primary cultures of cortical and spinal motor neurons obtained from SOD1G93A mouse models. In parallel, trimetazidine can increase autophagy in these cells. The authors also demonstrated that trimetazidine restores mitochondrial functionality of peripheral blood mononuclear cells from patients with sALS. The work is well designed and contributes to the understanding of pathogenesis and the development of novel disease-modifying therapy for ALS.

Comments:

1. Have other pathological processes that indicate the development of ALS pathology been identified in primary cell cultures? For example, the level of mutant SOD1 proteins in the cytoplasm, etc. What molecular mechanisms do the authors associate with the development of severe mitochondrial dysfunction (and not other manifestations of the pathology) at such an early stage? According to the literature, the progression of mitochondrial dysfunction in SOD1G93A mouse models begins to manifest itself after 4-6 months.

2. Was the purity of primary cultures of motor neurons derived from embryonic spinal cord and cortex determined in this work? The authors should provide data on the assessment of survival of motor neurons or MTT cell viability assay.

3. Figures 1, 2, and 3 show different levels of basal respiration for each of the WT and ALS groups in representative curves (the level of basal respiration in these figures for the group WT differs by 2 times). If all of these data were statistically processed together, would there still be a significant difference in basal respiration between these groups? Please explain.

4. The role of mitochondrial dyshomeostasis in the onset and development of ALS is being studied intensively. I recommend that the authors read some novel publications (PMID: 38002924, PMID: 38069154, PMID: 36111770, PMID: 35805131).

Comments on the Quality of English Language

Minor editing of English language is required

Reviewer 2 Report

Comments and Suggestions for Authors

The manuscript of Salvatori and co-workers describes the positive effects of Trimetazidine (TMZ) to ameliorate mitochondrial dysfunction in cell models of ALS. To study such effects, authors performed cell culturing, bioenergetic assays, electron microscopy and immunofluorescence as the main experimental approaches. The effects of TMZ were tested in different cell types and showed consistency. In most cases the data presented are correctly interpreted and discussed. Authors have a good collection of findings to support their main conclusions.

However, there are a couple of major issues I have found in the manuscript regarding bioenergetics, which should be addressed before this paper gets consideration for publication in the IJMS. I also include minor suggestions to improve the manuscript. I hope that my comments are helpful for the authors to generate a better version of this work.

-          Even though all oxygen consumption measurements were carried out by following the Seahorse protocol, I have noticed that the calculations of the bioenergetic parameters were incorrect in several cases. If I understood properly, such calculations were done using the Seahorse software and normalized using protein amount. For example, in Fig. 1, panels c-d, the reported “ATP production” respiration values were apparently the same as shown in “Basal respiration”. This can’t be true, since the respiration in WT cells is roughly 70 % lower than the basal respiration. In panel d, the “ATP production” value is even higher than the basal, which cannot be possible. In order to avoid this problem, please double-check manually all values in all figures showing these data. “ATP production” = [Basal respiration] – [Oligomycin-insensitive OCR (a.k.a. proton leak)]. Spare respiratory capacity = [FCCP-induced maximal respiration] – [Basal respiration]. I would also suggest changing “ATP production” to “ATP-linked respiration” because you are reporting OCRs not actual ATP as you do show in Fig 2e.f. As the non-mitochondrial residual OCR is considerable, please subtract it in all cases. To help visualize your data, I’d also suggest showing the y-axis in the same units of side-by-side panels to quickly compare them and see immediately the differences between PSC and PCC. In several cases, OCR units are shown with backslash (\) instead of forward slash (/), please amend accordingly.

-          In figure 2, the effect of TMZ is clear, but again the values of the different parameters are, on the one hand, probably miscalculated, and, on the other, normalized in a way that it was not straightforward for me to understand. The units of the y-axis are OCRs so I would have expected percentages or fractional numbers. To skip this issue, please report OCRs as in figure 1 and follow my suggestions for easier visualization.

-          In figure 3, a Seahorse analysis to evaluate the mitochondrial respiration promoted by substrates of complex I and later for complex II. The experiment is correctly performed, but the complex II/III dependent respiration should be called just “complex II dependent respiration” as in both cases for CI and CII, the electrons should be transferred to oxygen via the cytochrome pathway (CIII, Cyt c and CIV). Thus, to avoid confusion, please amend accordingly. Furthermore, authors added Ascorbate/TMPD at the end of the experiments, which did not promote CIV activity. Is there any idea why it didn’t work? Instead, data from spectrophotometric assays were collected but not shown in the paper. For scientific transparency, I would encourage the authors to include the data as supp. material.

-          Please check all calculations in Fig. 6, following the aforementioned points.

-          Authors should plot all bioenergetics data (histograms) as shown in Fig. S1, including all data points from the different replicates.

-          The main conclusion of the paper is that TMZ helps restore mitochondrial function, at least from the energy metabolism and morphological perspective, via a mechanism that involves autophagy activation. I don’t disagree with this possibility, but why not consider that biogenesis could also play a role here? In my opinion, the lower respiratory rates could reflect a lower mitochondrial biomass in the SOD1-G93A cells. Is there any data supporting that the mitochondrial content is similar? This could simply be tested by doing WB against mitochondrial proteins such as citrate synthase, OXPHOS subunits, VDAC, etc. It is actually mentioned in the methods that CIV activities were normalized using CS activities, perhaps you can check these data to answer this query. 

Reviewer 3 Report

Comments and Suggestions for Authors

The manuscript "Trimetazidine improves mitochondrial dysfunction in SOD1G93A cellular models of amyotrophic lateral sclerosis through autophagy activation" aimed to investigate the molecular pathways underlying the mechanism of action of TMZ in a neuronal context using primary spinal cord and cortical cell (PSC and PCC respectively) cultures from SOD1G93A mice.

The results show strong evidence that is consistent with the objective they sought to achieve, the authors showed that the acute administration of TMZ reverts SOD1G93A  phenotype improving respiration parameters, ATP production, electron transport chain (ETC) complexes activity and mitochondrial morphology.

The data are extensive and well presented, however, the quality of the image of OCR graphs should be improved. 

In contrast, the section Discussion should be improved by taking better advantage of the information provided by their data and also from evidence previously reported in the literature. As an example, the authors should discuss the lack of a positive control for autophagy in the cell culture treatments and the limitations of not having specific mitophagy protein markers such as PINK1 or parkin. Authors should also propose how TMZ could enhance the autophagy flux. A reference for a single TMZ concentration (10 μM) used for cell treatments should be also included.

Round 2

Reviewer 1 Report

Comments and Suggestions for Authors

The manuscript is appropriately revised and it can be published in the current state

Reviewer 2 Report

Comments and Suggestions for Authors

The authors have addressed all my comments. The paper is now ready for consideration.